

# Dry particle generation with a 3D printed fluidized bed generator

Michael Roesch[1] , Carolin Roesch[1] , and Daniel J. Cziczo[1,2]

[1]Department of Earth, Atmospheric & Planetary Sciences, Massachusetts Institute of Technology, Cambridge, 02139, USA
[2]Department of Civil Environmental Engineering, Massachusetts Institute of Technology, Cambridge, 02139, USA

*Correspondence to*: Michael Roesch (roesch@mit.edu)

**Abstract.** Here we describe the design and testing of a compact fluidized bed aerosol generator named PRIZE (PRinted fluidIZed bed gEnerator) manufactured using stereolithography (SLA) printing. Dispersing small quantities of powdered materials – due either to rarity or expense - is challenging due to a lack of small, low-cost dry aerosol generators. With this as motivation, we designed and built a generator that uses a mineral dust or other dry powder sample mixed with bronze

beads that sit atop a porous screen. A particle free airflow is introduced, dispersing the sample as airborne particles. Particle number concentration and size distributions were measured during different stages of the assembling process to show that the SLA 3D printed generator did not generate particles until the mineral dust sample was introduced. Time-series measurements with Arizona Test Dust (ATD) showed stable total particle number concentrations of 10 – 150 cm$^{-3}$, depending on the sample mass, from the sub- to super-micrometer size range. PRIZE is simple to assemble, easy to clean,

inexpensive and deployable for laboratory and field studies that require dry particle generation.

## 1 Introduction

Investigating dry powder samples such as mineral and soil dust and volcanic ash is essential to understand their atmospheric influence, especially on clouds (Boucher et al., 2013). Inexpensive commercial nebulizers have often been used to aerosolize

these types of samples but require they first be made into a water slurry. Garimella et al. (2014) demonstrated that such aqueous processing alters the surface composition and hygroscopicity of the particles even after condensed phase water is completely removed. These samples therefore need to be prepared with suitable particle generators that do not change their characteristics. Different types of generators such as rotating brush generators (Cziczo et al., 2013, Hiranuma et al., 2015), fluidized bed generators (FBG) (Tobo et al., 2012, Hartmann et al., 2016) and shakers (Garimella et al., 2014) have been

used for dry dispersion depending on the amount of material and the experimental setup. In some applications only a small amount of aerosol particles are needed or for a short time period. These include filter sampling (Dryer at el., 2014), particle trapping (Hesse et al., 2002) and single particle mass spectrometry (Murphy, 2006). Moreover, the sample to be aerosolized might be limited, due to rarity or expense, necessitating a generation system capable of working with gram-level quantities. These uses motivate a small, low-cost and easy to set up dry particle generator.





Fluidized bed aerosol generators (FBGs) have been used in several studies to disperse dry samples as sub- to super-micrometer size range aerosol particles (Guichard, 1976; Moreno et al., 1976; Boucher and Lua, 1982; Wang et al., 1998; Gauthier et al., 1999; Niedermeier et al., 2010; Clemente et al., 2013) and are commercially available e.g. Fluidized Bed Aerosol Generator (FBAG, Model 3400A, TSI Inc.). Small particles, sub-10 micrometers, are difficult to disperse due to

cohesive forces (Geldart, 1973); the dispersibility of a dust sample increases with increasing particle size (Hinds, 1999). While a broad size range of the material being dispersed typically enables better aerosolization it may also lead to particle segregation in the bed (Lind et al., 2010). Another issue faced by FBGs is that the material to be dispersed is often charged due to triboelectrification. This is caused by friction between the generator walls, the beads and the particles themselves (Mehrani et al., 2007). Using conductive wall material or ionization through radioactive neutralizers decreases this effect

(Boucher and Lua, 1982; Forsyth et al., 1998).

An alternative means of particle production is agitation of dry materials (Sullivan et al., 2009). Examples include flasks in combination with ultrasonic bath or mechanical shakers to disperse e.g. calcite powder as described by Sullivan et al., 2009. These techniques require multiple instruments and supervision of the generation setup.

The objective of this study was to produce a small 3D printed FBG able to disperse small quantities of dry micrometer-sized

samples without artifact particles from either the generator or bed, which will enable groups to use a comparable aerosol generation method. In comparison to the FBAG or the small-scale powder disperser (SSPD, Model 3433, TSI Inc.) the PRIZE generator does not contain any mechanical moving parts, features smaller dimensions, weight, much lower acquisition costs and maintenance.

## 2 Methods


### 2.1 Design

PRIZE consists of two main parts, the generator body (Fig. 1) and exchangeable lids with customizable outlet configurations (Fig. 2). The instrument, except the porous screen and the bronze beads comprising the bed, was designed using a computer aided design (CAD) program (Solidworks 2015, Dassault Systems). The PRIZE generator follows similar designs by Marple

et al., 1978. But, in contrast to the aforementioned generator or the commercially available FBAG and SSPD, PRIZE does not contain any moving parts or a supply chain to feed fresh dry powder into the bed. This feature was chosen to keep the set up simple and reproducible.

At the bottom of the generator there is a 6.35 mm (0.25 inch) tube inlet for introduction of a particle free carrier gas. The inlet was designed to fit common tube connectors. Downstream of the inlet, an internal cone with 7 radially equidistant arms

support a 25.4 mm diameter porous screen. This screen supports the bed and sample while also allowing a homogeneous flow pattern through the bed. The porous screen is made from stainless steel (TWP Inc., CA) with a mesh size of 80 micrometers. This prevents fall through of the 100 micrometer diameter bronze beads that form the bed. Above the porous



screen is a 35 mm long and 25.4 mm diameter elutriator tube; the bronze bead bed is located inside the elutriator. The elutriator tube is topped with a lid with 6.35 mm (0.25 inch) outlets. To prevent leakage, lids are equipped with inner side O-rings. Based on the number of instruments connected to the generator, lids designed with single or multiple 6.35 mm outlets can be installed (a single outlet lid is shown in Fig. 2). Mounts for a rotameter and a 3 mm LED to illuminate the bed were

designed at the left side of the generator body. Mounting holes at the bottom were designed to secure the generator to a surface during operation. The overall dimensions of PRIZE are 70 mm × 60 mm × 98 mm (width, depth, height). All designed parts are saved as style files (.stl) to be readable by the 3D printer software.

## 2.2 Manufacturing

The generated construction files were uploaded to the 3D printer software (PreForm, Formlabs Inc.). In the program, the

parts were oriented on the virtual build platform and scaffolding with 0.5 mm contact points were generated for support during the printing process. The oriented and supported parts were then positioned on the virtual build platform and uploaded to the 3D SLA printer (Form 2, Formlabs Inc.). Clear photopolymer resin (FLGPCL02, Formlabs Inc.) was used as the printing material; this allowed observation of particle production when PRIZE was lit by the mounted LED. At the start of printing the resin was automatically heated to 31°C and kept at this temperature throughout the process. The liquid resin is

cured through photo polymerization by a 405 nm violet laser. The resolution of the printed layers can be adjusted to 25, 50 or 100 micrometer. Using the highest resolution, of 25 micrometer, printing was ~19 hours while using the lowest resolution only ~7 hours were required. The PRIZE generator results considered here was printed with 100 micrometer resolution.
The printed parts were removed from the build platform and agitated in isopropyl alcohol (IPA) for ~20 minutes to remove uncured resin. Snips were used to remove the support structure from the printed parts. Residual supports marks were

removed by wet sanding in a two-step process: first coarse sanding with grit size 800 and a second fine sanding with grit size 2000. To increase the strength of the printed parts, the manufacturer recommends a 60 minute post-curing with a 405 nm light source (it should be noted this curing time also depends on the size and the wall thickness of the printed part). The printed generator parts were post-cured overnight (~8 hours) in a custom-built UV box. Inside are 300 surface-mount device (SMD) light-emitting diodes (LEDs) emitting at 405 nm, with a total intensity of 934.8 cd. After post-curing the parts were

polished using a Dremel with a soft felt bob.

### 2.3 Experimental setup
A schematic of the experimental setup with the relevant flow rates used in this study is shown in Fig. 3. Dry filtered nitrogen

was used as the carrier gas. The flow was controlled by a rotameter (MR3A, Omega Engineering). Flow tests with only the 100 micrometer bronze beads (ACuPowder International LLC) showed that 4.0 l min$^{-1}$ was sufficient to create a "boiling" motion in the fluidized bed (i.e., the threshold where the gas did not only pass through the pore space of the beads without moving them). All measurements in this study were therefore performed at 4.0 l min$^{-1}$. At the outlet of the generator the flow





was split into three pathways with equal tubing length. The first was connected to a condensation particle counter to measure particle number concentrations in the size range from 0.007 to 2.0 micrometer (CPC, BMI Inc.), the second to an optical particle sizer to determine particle number size distributions in the size range from 0.3 to 10 micrometer (OPS Model 3330, TSI Inc.) and the third to a filter open to lab for excess air.

## 3. Results

Three experiments were performed to demonstrate PRIZE performance. First, control experiments at different assembly stages to verify minimal particle generation by the generator itself. Second, a mineral dust sample was added to the generator
and a time-series measurement was performed. Third, a sensitivity study was performed on the effect of generated particle number concentration as function of mineral dust mass added to the generator.

### 3.1 Control experiments

For the first control experiment, the CPC and OPS were run for 600 seconds with an upstream filter (IDN-4G, Parker) at
their inlet to verify zero counts were recorded (Fig. 4a, Fig. 5a). The CPC and OPS were then connected to the generator as shown in Figure 3. Measurements at different stages of the assembling process of the generator were performed to verify minimal particle production (Table 1). The out of printer stage did not show particles being generated by the instrument, resulting in zero counts in the CPC and OPS. After sanding the generator, particle concentrations up to 60 cm$^{-3}$ were detected in the CPC during the first 60 seconds of the measurement but asymptotes to zero by 450 seconds; afterwards no particles
were detected in the OPS. Introducing the porous screen did not result in further particle generation (i.e., only four single CPC counts in 600 seconds). Particles were detected by the CPC and the OPS after introducing the bronze beads into the generator. The averaged size distribution of the OPS measurements showed a maximum particle concentration of ~1.5 cm$^{-3}$ with a mode at ~2.5 micrometers diameter. Similar particle number concentrations were detected by the CPC.

### 3.2. Mineral dust experiments

Arizona Test Dust (ATD, Powder Technology Inc. MN) was used as the sample in this study. The nominal size of particles ranged from 0-3 micrometer. 0.2 g ATD was added to the generator. A measurement time of 1200 seconds was split into two 600 second segments (Fig. 4 and Fig. 5) to determine the particle concentration as a function of time. The first 600 seconds showed a decrease after startup, a brief drop in concentration, and then an upward asymptote to ~25 cm$^{-3}$ at the CPC (Fig.
6a). The size distribution in the OPS remained more or less constant with a final maximum particle concentration of ~25 cm$^{-3}$ centered at 0.3 micrometer diameter (Fig. 6b). Although this is the lowest size bin of the OPS, the similarity in CPC and OPS measurements indicate most generated particles fall within the OPS size range. The second 600 seconds showed a stable particle concentration from 25−30 cm$^{-3}$ in the CPC (Fig. 6c). In the OPS, the shape of the size distribution did not change over time, showing the maximum particle concentration at ~0.3 micrometer (Fig. 6d).



A sensitivity study of the number concentration of generated particles as a function of ATD mass added to the generator was then performed. ATD was added stepwise from 0.1 to 0.5 g to the clean PRIZE instrument and it was run for 600 seconds, the stabilization time indicated in the initial experiment. This procedure was repeated at each mass loading. The resulting particle number concentrations showed an exponential growth with increasing mass load in the fluidized bed from $10 - 150$

$cm^{-3}$ (Fig. 7).

### 4. Discussion

Before generator assembly, time-series of the CPC and OPS were performed. No particles were detected, providing a zero background for the experiments. Connecting the generator to the CPC and OPS after printing also showed no particles,

indicating that PRIZE was not generating particles at this stage. Particles observed by the CPC post-sanding are likely residuals of the sanding process; we suggest a thorough cleaning (e.g. via immersion sonication) to eliminate these although they were observed to asymptote to zero after 450 seconds. Because no particles were observed in the OPS, we assume their diameter was smaller than 0.3 micrometer (i.e., below the detection limit of the OPS). Installing the porous screen into the generator likewise did not cause significant particle generation. Introduction of the bronze beads resulted in an OPS

maximum concentration of ~1.5 $cm^{-3}$ with a mode at ~2.5 micrometers diameter; a similar particle concentration was detected at the CPC.

The mineral dust experiment, using ATD as the sample, were split into two parts. In the first 600 seconds, a "warm-up" period of the generator is visible with transient and increasing particle concentration at the CPC. The particle size distribution in the OPS showed only minor variations over the first 600 seconds. The maximum concentration was detected

at 0.3 micrometers, the lowest bin of the OPS. The second 600 seconds of the measurement showed a stable output of the generator with concentration values $25-30$ $cm^{-3}$ at the CPC. The OPS size distribution did not change over time, maintaining a maximum concentration at 0.3 micrometer. We conclude that the similarity of CPC and OPC concentrations means the size range of aerosol particles generated from the ATD sample was from 0.3 to 2.5 micrometer with the highest abundance at the lower end of this range. A sensitivity study of generated particle number concentration as function of ATD mass added to the

fluidized bed showed an exponential behavior with increasing mass load from 10 to 150 $cm^{-3}$ for 0.1 to 0.5 g.

We expect that the size range of generated aerosol particles will depend on the properties of the dispersed dry material with these results specific to this ATD sample. We suggest that the PRIZE be operated for ~600 seconds before measurement to avoid the transient period.

### 5. Conclusion

This study describes the design, manufacture and proof-of-concept of a 3D printed FBG called PRIZE. The described generator is capable of dispersing aerosol particles from dry material without itself generating significant particles (~5% by number at 0.2 g of ATD). Using SLA technology makes this a low-cost instrument when compared to commercially



available FBGs. Furthermore, the generator is compact and easy to set up. Requiring only particle-free pressurized gas, it is ideal for use in minimally appointed laboratory and field conditions where dry particle dispersal is required.

The PRIZE FBG can be used in future studies for investigating the chemical composition of different dust species, e.g. soil dust, fly ash or other mineral dust, with mass spectrometry or transmission electron microscopy. Due to the preservation of

the original chemical composition of the particles, which is a major advantage of dry particle generation, the aerosolized particles can be investigated with regards to their water droplet and ice nucleating potential.

The .stl files for PRIZE are available up on request.

**Conflict of interest**

The authors declare that they have no conflict of interest.

**Acknowledgements**

We acknowledge funding from the Center for Complex Engineering Systems at the King Abdulaziz City for Science and Technology, the Massachusetts Institute of Technology, NSF (AGS-1461347), DOE (DE-SC0014487), and NASA

(NNX13AO15G) for funding.

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



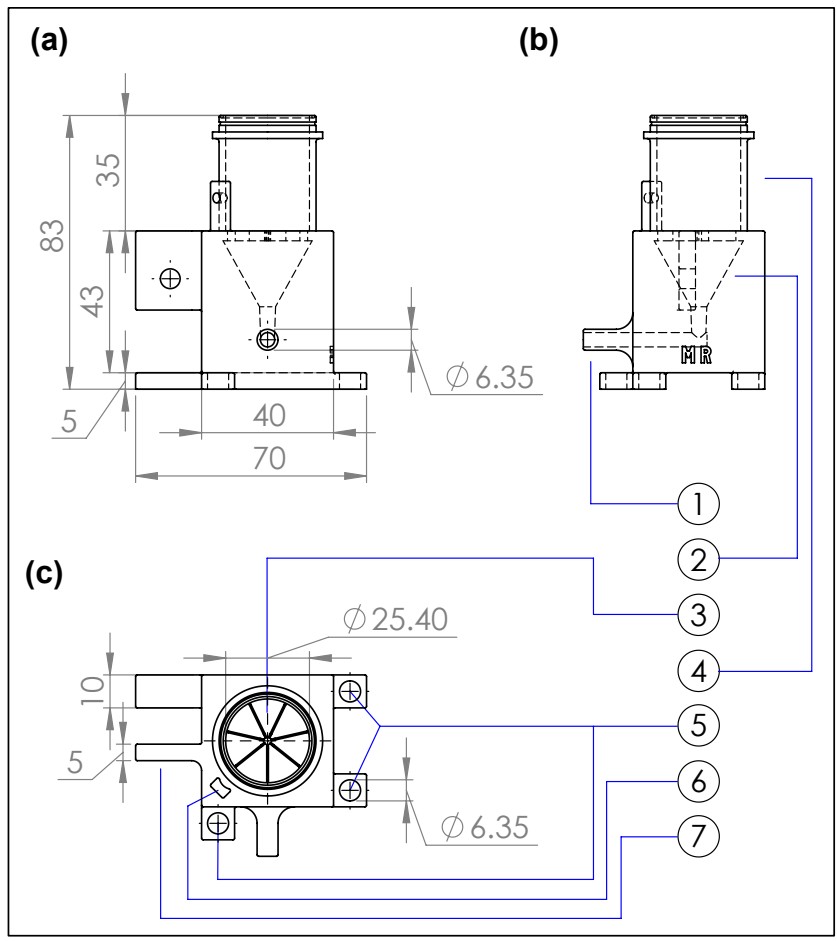

**Figure 1:** Dimensioned drawings of the PRIZE body: (a) rear view; (b) side view; (c) top view. All dimensions are in millimeters. The generator body consists of 7 components; (1) inlet, (2) internal cone, (3) radial grid with openings, (4) elutriator tube, (5) mounting holes, (6) LED mount, and (7) rotameter (or other flow meter) mount.



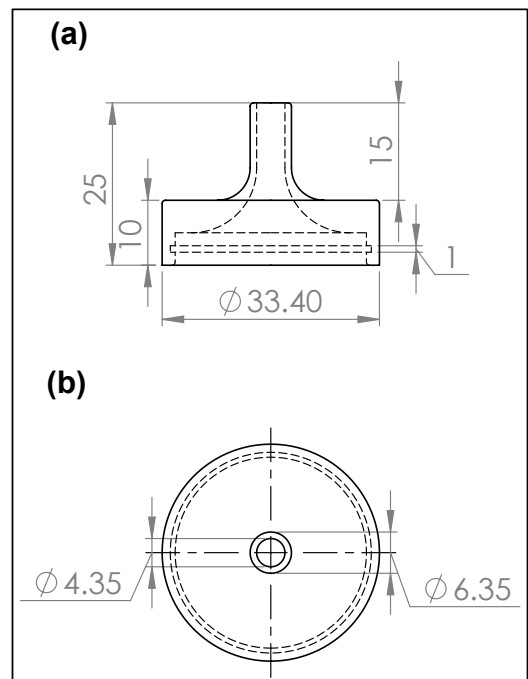

**Figure 2: Dimensioned drawing of a single port outlet lid: (a) side view; (b) top view. All dimensions are in millimeters.**

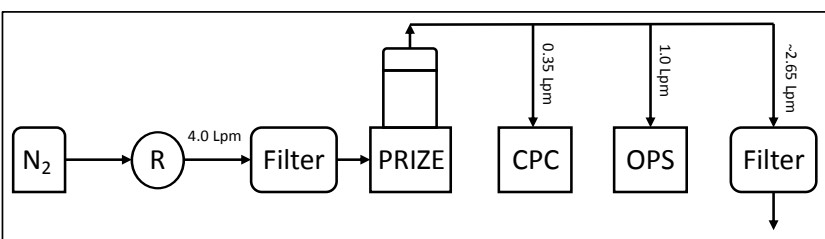

5    **Figure 3: Schematic of the experimental setup used in this study. A filtered dry nitrogen flow was controlled by a rotameter (R) upstream of PRIZE. Downstream, the flow was split into three equal length pathways to a CPC, an OPS, and excess flow discarded through a filter.**





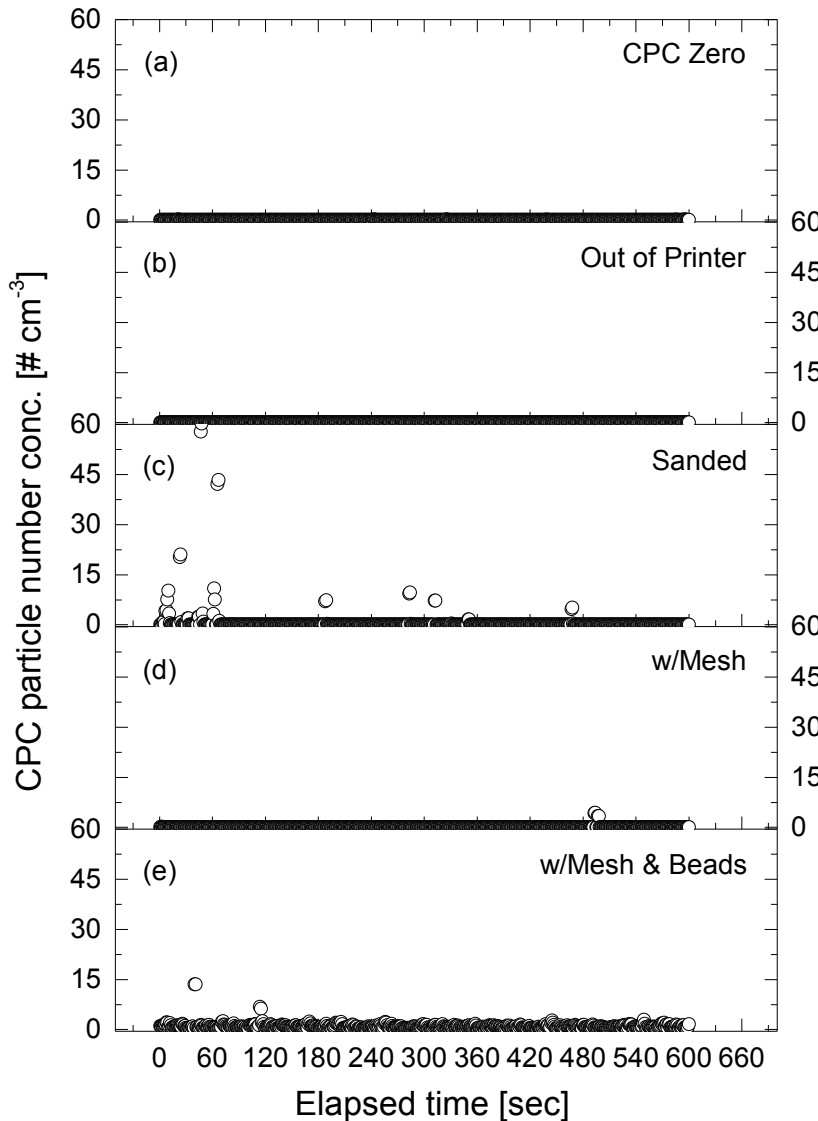

Figure 4: Time-series measurements of particle number concentrations: (a) CPC filter measurement; (b-e) at different assembling stages of the PRIZE generator.





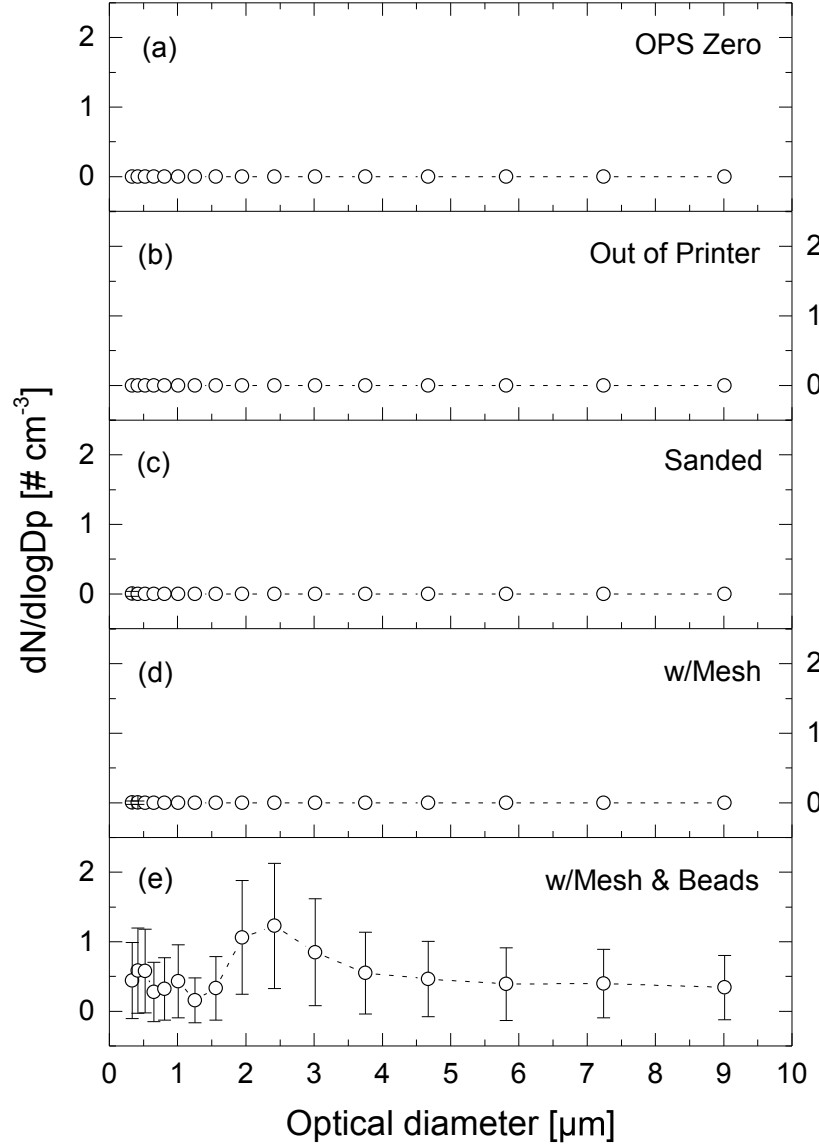

**Figure 5:** Averaged particle number size distributions: (a) OPS filter measurement; (b-e) at different assembling stages of the PRIZE generator.



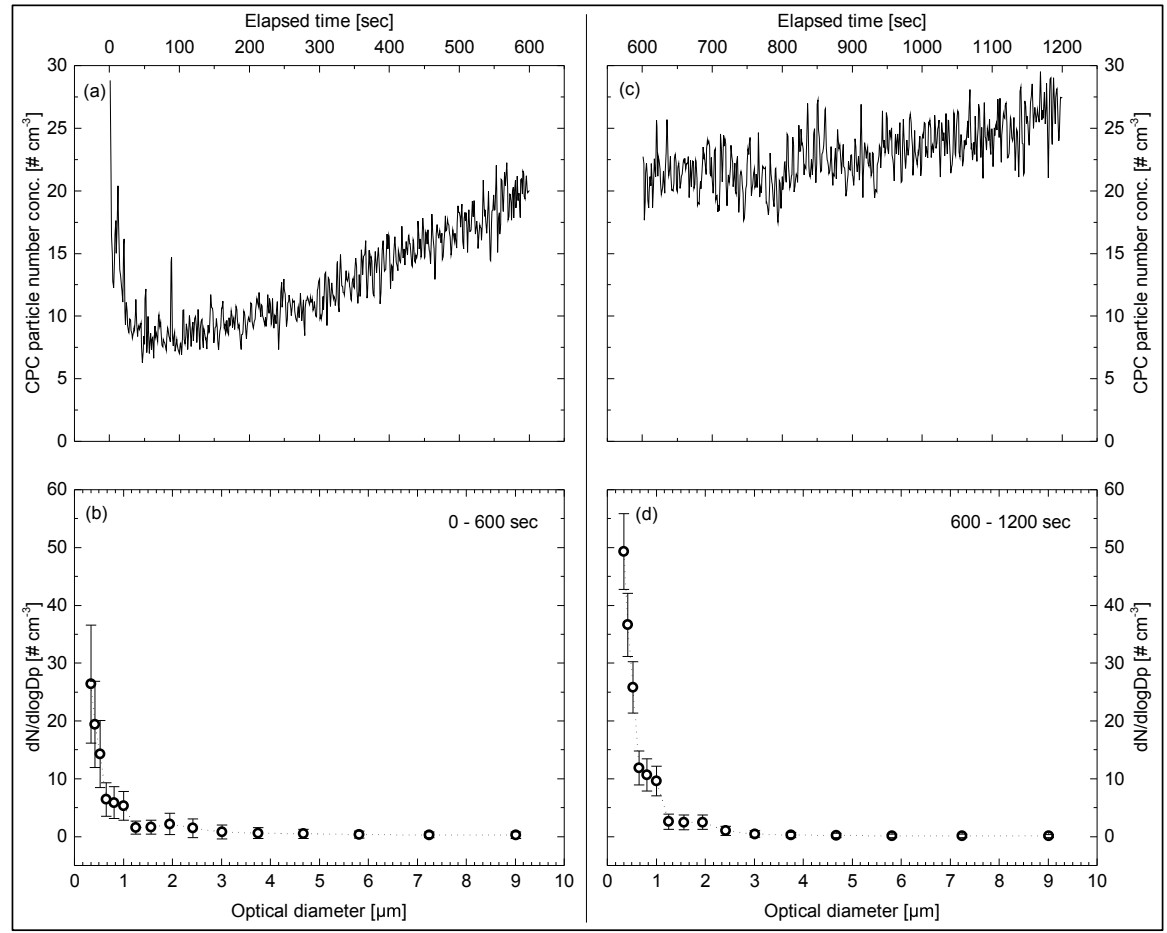

**Figure 6: Arizona Test Dust measurements: (a) particle number concentration for the first 600 seconds; (b) corresponding averaged particle number size distribution; (c) particle number concentration for the second 600 seconds; (d) corresponding averaged particle number size distribution.**




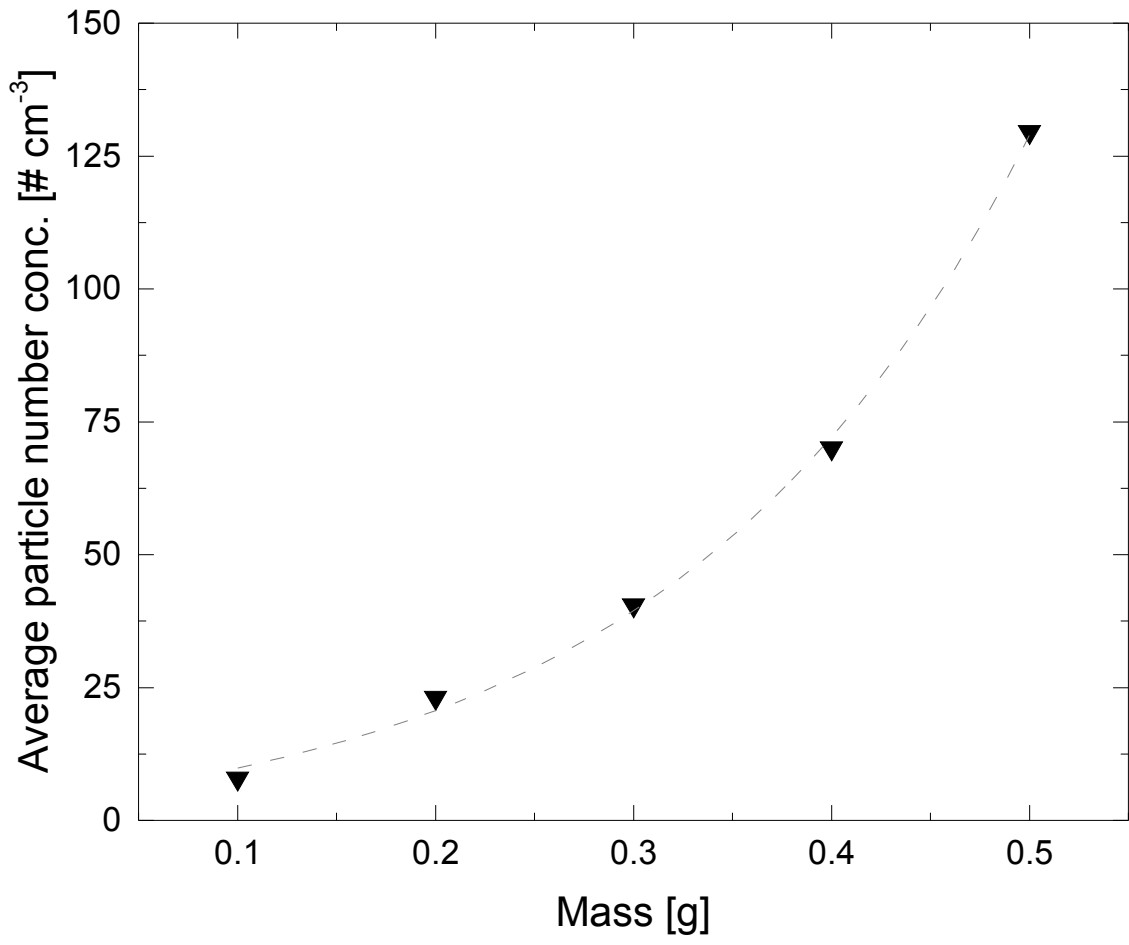

**Figure 7: Average particle number concentration as function of Arizona Test Dust mass in the fluidized bed (triangles) with a fitted exponential curve (dashed line).**



**Table 1: Assembly stages of the PRIZE generator, the action which was taken and the particle detection results at the CPC and OPS. Corresponding particle number concentration time-series and averaged particle size distributions can be found in Figures 4 and 5, respectively.**

| Stage | Action | CPC - particles | OPS - particles |
|-------|--------|-----------------|-----------------|
| **Out of printer** | Raw print tested | No (4b) | No (5b) |
| **Sanded** | Raw print wet sanded with 800 & 2000 grit | Yes (4c) | No (5c) |
| **w/Mesh** | Porous screen was installed into the sanded generator | Yes (4d) | No (5d) |
| **w/Mesh & Beads** | Bronze beads were added to the bed | Yes (4e) | Yes (5e) |