# Peer review of "Dry particle generation with a 3D printed fluidized bed generator"

_Atmospheric Measurement Techniques, 2016_

## Referee Comment (RC1) · Anonymous Referee #1 · 2 Feb 2017

Generating aerosol particles via dry dispersion is a non-trivial task in aerosol research, though a number of techniques have been proposed and used. Roesch et al. have described and developed a simple 3D printed fluidized bed generator, and evaluated its performance by dispersing Arizona Test Dust. This instrument can be very useful in mineral dust aerosol research. I would recommend it for final publication after the following comments are properly addressed.

**Major comments:**

The generator can generate aerosol with total concentrations up to ~150 cm$^{-3}$ if 0.5 g ATD sample is added into the generator. For aerosol research, monodisperse particles are typically required. Therefore, if the aerosol flow exiting the generator is passed through a DMA to produce monodisperse aerosol, the resulted aerosol concentration might be very low. This may largely limit the applicability of this generator.

**Minor comments:**

Page 1, line 20-23: In addition to Garimella et al. (2014), there are a number of studies showing that wet generation could change the physicochemical properties of mineral dust particles, including Sullivan et al. (2010) and Kumar et al. (2011). This issue has also been discussed by a recent review paper (Tang et al., 2016).

Page 2, line 11-14: I do not agree with the statement "These techniques require multiple instruments and supervision of the generation setup". As pointed out in the same paragraph, those generators are actually very simple and only require a flask and a shaker. I believe the major drawback is that particle number concentration typically shows fast and large variation (e.g., Tang et al., 2015).

Figures 4-5: Could authors briefly explain in figure captions what each stages are actually doing? This will help readers better understand these two figures.

**Technical comments:**

I suggest authors should carefully check the entire manuscript as there are quite a lot of places where typos and grammatical issues occur. Below I only provide a few examples:

Page 1, line 8: should it be "due to either rarity or expense"?

Page 1, line 20: change "require they first be made…" to "require them first to be made…"?

Page 1, line 26: change "needed or for…" to "needed for…"?

Page 2, line 25: delete "But" or change it to "However"?

Page 3, line 17: the last sentence in this paragraph is grammatically incorrect.

Page 5, line 8: The first sentence is awkward. We do not perform time-series. What we can perform is measurement.

Page 5, line 24: change "as function" to "as a function".

Page 6, line 6: should the more professional terminology be "cloud condensation nucleation and ice nucleation potential"?

**References:**

Kumar, P., Sokolik, I. N., and Nenes, A.: Cloud condensation nuclei activity and droplet activation kinetics of wet processed regional dust samples and minerals, Atmos. Chem. Phys., 11, 8661-8676, 2011.

Sullivan, R. C., Moore, M. J. K., Petters, M. D., Kreidenweis, S. M., Qafoku, O., Laskin, A., Roberts, G. C., and Prather, K. A.: Impact of Particle Generation Method on the Apparent Hygroscopicity of Insoluble Mineral Particles, Aerosol Sci. Technol., 44, 830-846, 2010.

Tang, M. J., Whitehead, J., Davidson, N. M., Pope, F. D., Alfarra, M. R., McFiggans, G., and Kalberer, M.: Cloud Condensation Nucleation Activities of Calcium Carbonate and its Atmospheric Ageing Products, Phys. Chem. Chem. Phys., 17, 32194-32203, 2015.

Tang, M. J., Cziczo, D. J., and Grassian, V. H.: Interactions of Water with Mineral Dust Aerosol: Water Adsorption, Hygroscopicity, Cloud Condensation and Ice Nucleation, Chem. Rev., 116, 4205–4259, 2016.

---

## Referee Comment (RC2) · Anonymous Referee #2 · 9 Mar 2017

Authors describe a new design to generate small quantities of powder material. There are several techniques available in the market but the new design is slightly different and unique particularly as authors claim that it is small and low-cost. The characterization performance results are convincing, however, and this is optional if authors could also present some research application results. Major comment is it is not clear 'what are the limitations of the existing aerosol generators.' It is mentioned that (page 2 line 13) the existing flask design requires multiple instruments and supervision of the setup. This is incorrect. Further, they say FBAG and SSPD involves mechanical moving parts and larger weight. It is not clear how having multiple moving parts and weight (< 50 lbs) impedes the research ability of the instrument to generate the dry powder. Some detailed discussion on this topic would be very useful.

---

## Author Comment (AC1) · 28 Apr 2017

**Referee #1,**

We thank you for your thoughtful comments and thorough reading of the paper. We have made the suggested changes and added data and references where requested. A point by point response follows.

**Major comments:**

**1. The generator can generate aerosol with total concentrations up to ~150 cm-3 if 0.5 g ATD sample is added into the generator. For aerosol research, monodisperse particles are typically required. Therefore, if the aerosol flow exiting the generator is passed through a DMA to produce monodisperse aerosol, the resulted aerosol concentration might be very low. This may largely limit the applicability of this generator.**

We agree that monodisperse particles are essential for size resolved measurements and calibrations. In this study our focus was particle generation for applications with low material or particle requirements. However, based on this comment we have performed a particle number calculation as a function of mineral dust load to estimate the total number concentration for larger dust loads than 0.5g in the generator. Using the equation derived from the fitted curve to our measurements (y = 8.414*exp(x/0.18081)-4.75126) for this batch of ATD, a dust load of 0.75g will ideally generate ~528 cm$^{-3}$, a dust load of 1g ~2118 cm$^{-3}$ and 1.25g already 8456 cm$^{-3}$. Introducing these concentrations into a DMA, and assuming 10% of the introduced particles are selected as monodisperse aerosol particles, output concentrations of ~53 cm$^{-3}$ – 845 cm$^{-3}$ are achievable with the final output concentration of monodisperse particles dependent on several other factors e.g. particle size, shape etc. (Wiedensohler et al. 2012).

In the manuscript we added the following text per this comment:

(Page 5, line 14-24): added

"A curve fitted to the data provides a particle number concentration (PNC) as a function of the mass load (ML) for this ATD sample:

$$PNC = 8.414 * \exp\left(\frac{ML}{0.18081}\right) - 4.75126 \qquad\qquad \text{(Eq. 1)}$$

Eq. 1 can be used to estimate the generated particle number concentration. For a sample mass of 0.75 g, ~525 cm$^{-3}$; 1 g ~2120 cm$^{-3}$ will ideally be generated. The exponential form of Eq. 1 should not be used to multi-gram quantities; it is used here to demonstrate that particle size selection instruments e.g. a differential mobility analyzer (DMA), could be used in combination with PRIZE and higher mass loadings. While the purpose of this work is to demonstrate the

applicability of PRIZE for the aerosolization of small sample sizes, some researchers may use it for dispersion followed by separation. Assuming ~10% of the introduced particles are selected as monodisperse aerosol particles, output concentrations of ~15 cm$^{-3}$ - 210 cm$^{-3}$ are achievable. The final output concentration of monodisperse particles will depend on several factors, including the sample material mode size, the particle size selected, the shape factor, etc. (Wiedensohler et al., 2012)."

(Page 6, line 8): added

"Furthermore, we demonstrate with calculations that mass loadings larger than 0.5 g could be used in combination with differential mobility separation for production of size-selected aerosols."

Also applicable to this point is that an increase in the amount of particles produced would increase a size-selected fraction. To address this we performed experiments with other PRIZE configurations, including a stainless steel insert. For this we introduced a stainless steel tube (25.4mm in diameter, 20mm in height and 2mm wall thickness) inside the elutriator to prevent the direct contact of the bronze beads to the wall to improve generation and minimize direct particle contact to the 3D printed material. This feature reduced the particle number concentration generated from the wall material down to ~0 particles cm$^{-3}$ (in comparison to ~1.5 particles cm$^{-3}$ without the stainless steel insert).

We have also added the following text and modified Figures 4e and 5e per this comment:

(Page 3, line 5): added

"We investigate configurations both with and without a stainless steel insert forming the walls of the elutriator."

(Page 4, line 29): added

"The particle concentration at this stage could be reduced to ~0.1 cm$^{-3}$ with insertion of a stainless steel tube into the elutriator (Fig. 4e, Fig. 5e). This prevented direct contact of the bronze beads with the generator wall and indicates that some abrasion of the printed surface can take place."

(Figure 4e):
[Figure]

(Figure 5e):

[Figure]

**Minor comments:**

**1.      (Page 1, line 20-23): In addition to Garimella et al. (2014), there are a number of studies showing that wet generation could change the physicochemical properties of mineral dust particles, including Sullivan et al. (2010) and Kumar et al. (2011). This issue has also been discussed by a recent review paper (Tang et al., 2016).**

We thank the reviewer and included the additional suggested literature on wet generation of mineral dust particles. We also thank the referee for the additional literature references in the review; we also added them to the correct position in the manuscript.

now (Page 1, line 21-24):

"Sullivan et al. (2010), Kumar et al. (2011), Garimella et al. (2014) and Tang et al. (2016) demonstrated that such aqueous processing alters the surface composition, physiochemical properties and hygroscopicity of the particles even after condensed phase water is completely removed."

**2.      (Page 2, line 11-14): I do not agree with the statement "These techniques require multiple instruments and supervision of the generation setup". As pointed out in the same paragraph, those generators are actually very simple and only require a flask and a shaker. I believe the major drawback is that particle number concentration typically shows fast and large variation (e.g., Tang et al., 2015).**

We agree. The statement "These techniques require multiple instruments and supervision of the generation setup." in the manuscript on page 2, line 13 was deleted.

**3.    Figures 4-5: Could authors briefly explain in figure captions what each stages are actually doing? This will help readers better understand these two figures.**

We included a more detailed description of the different stages in both figure captions.

(Figure 4):

"Figure 4: Time-series measurements of particle number concentrations: (a) CPC with filter; (b) freshly printed generator; (c) through the generator after wet sanding; (d) through the generator after wet sanding and installing the porous screen; (e) fully assembled generator including porous screen and bronze beads atop as bed material, without (solid line) and with a stainless steel insert (dashed line)."

(Figure 5):

"Figure 5: Average particle size distributions: (a) OPS with filter; (b) freshly printed generator; (c) through the generator after wet sanding; (d) through the generator after wet sanding and installing the porous screen; (e) fully assembled generator including porous screen and bronze beads atop as bed material, without (circles) and with a stainless steel insert (triangles)."

**Technical comments:**

**"I suggest authors should carefully check the entire manuscript as there are quite a lot of places where typos and grammatical issues occur. Below I only provide a few examples:"**

We thank the referee and have now performed a thorough check of grammar and typos. Please find below the corrections that have been made to the manuscript. All changes that have been made in addition to the comments can be seen in the trackable version of the manuscript.

1.    "Page 1, line 8: should it be "due to either rarity or expense"?

(Page 1, line 8): changed to
"due to either rarity or expense"

2.      "Page 1, line 20: change "require they first be made…" to "require them first to be made…"?"

now (Page 1, line 21): changed to
"require them first to be made.."

3.      "Page 1, line 26: change "needed or for…" to "needed for…"?"

now (Page 1, line 28): changed to
"is needed, or for a short period of time."

4.      "Page 2, line 25: delete "But" or change it to "However"?"

now (Page 2, line 29): deleted
"But"

5.      "Page 3, line 17: the last sentence in this paragraph is grammatically incorrect "

now (Page 3, line 21): sentence was changed to
"The PRIZE generator used in this study was printed with 100 micrometer resolution with no significant performance changes observed across this resolution range."

6.      "Page 5, line 8: The first sentence is awkward. We do not perform time-series. What we can perform is measurement."

now (Page 4, line 21): changed sentence to
"Before generator assembly, measurements with the CPC and OPS coupled to a filter (IDN-4G, Parker) were performed."

7.      "Page 5, line 24: change "as function" to "as a function"."

now (Page 5, line 10): changed to
"as a function"

8.      "Page 6, line 6: should the more professional terminology be "cloud condensation nucleation and ice nucleation potential"?"

(Page 6, line 6): changed the sentence to
"Due to the preservation of the original chemical composition of the aerosolized particles, which is a major advantage of dry particle generation, investigations of cloud condensation and ice nucleation potential can be made without aqueous processing artifacts."

---

## Author Comment (AC2) · 28 Apr 2017

**Referee #2,**

We thank you for your thoughtful comments and thorough reading of the paper. We have made the suggested changes and added data and references where requested. A point by point response follows.

**Major comments:**

**1.    The characterization performance results are convincing, however, and this is optional if authors could also present some research application results.**

Since submission of this "proof of concept" of the printed fluidized bed generator we have now put it into operational use to generate mineral dust particles in our lab. Specifically, we are studying the effects of mineral dust particles on the output of different solar cells.

Towards this point we have added the following text and a figure (Fig. 8) of the generated particle number size distribution:

(Page 1, line 14):

"Additional tests with collected soil dust samples are also presented."

(Page 4, line 16-19):

"In addition, PRIZE was used to disperse an arid soil sample collected in Saudi Arabia. Data for each experiment and the soil dust dispersion are presented in the subsequent sections."

(Page 5, line 26-29):

"A final experiment was conducted to demonstrate the use of PRIZE for dispersion of collected soil dust samples. Fig. 8 provides a size distribution of particles dispersed from an arid soil sample collected in Dhahrat Laban (west of Riyadh, Saudi Arabia)."

(Page 6, line 5):

"We demonstrate the use of PRIZE for collected samples of soil dust and note its use with mass spectrometry or transmission electron microscopy. "

(Page 16, line 1):

Figure 8:

[Figure]

**2.    It is not clear 'what are the limitations of the existing aerosol generators.'**

The presented particle generator is a low-cost addition to already existing dry particle generation instruments. To expand on this we added the following statement to the manuscript.

(Page 5, line 31):

"This study describes the design, manufacture and proof-of-concept experiments of the 3D printed fluidized bed generator PRIZE, which is a compact, simple and low-cost addition to existing dry particle generation instruments. "

**3.    It is mentioned that (page 2 line13) the existing flask design requires multiple instruments and supervision of the setup. This is incorrect.**

We agree. The statement on page 2, line 13 was deleted.

**4.    Further, they say FBAG and SSPD involves mechanical moving parts and larger weight. It is not clear how having multiple moving parts and weight (< 50lbs) impedes the research ability of the instrument to generate the dry powder.**

We agree that the weight of an instrument has no influence on the ability to generate particles from dry powder. However, a lower weight of the instrument would further enable its use in e.g. remote field applications. To clarify this we added the following statement to the manuscript.

(Page 2, line 20):

"In comparison to existing dispersion devices, the PRIZE generator does not contain moving parts, features smaller dimensions and mass, and has a lower cost, requiring only access to 3D printing. This allows for multiple PRIZE generators to be used with different samples, thereby reducing the time and possible artifacts associated with cleaning procedures on a single generator."